# Optimal Rotational Angular Velocity Determination Method Based on Compound Rotary Semi-Strapdown Inertial Navigation System

**DOI:** 10.3390/s22124583

**Published:** 2022-06-17

**Authors:** Chenming Zhang, Jie Li, Xiaoqiao Yuan, Xi Zhang, Xiaokai Wei, Kaiqiang Feng, Chenjun Hu, Debiao Zhang, Yubing Jiao

**Affiliations:** 1National Key Laboratory for Electronic Measurement Technology, North University of China, Taiyuan 030051, China; sf20210601@st.nuc.edu.cn (C.Z.); s1906172@st.nuc.edu.cn (X.Y.); b1806023@st.nuc.edu.cn (X.W.); b1506011@st.nuc.edu.cn (K.F.); b20210602@st.nuc.edu.cn (C.H.); s2006029@st.nuc.edu.cn (Y.J.); 2School of Electrical Control Engineering, North University of China, Taiyuan 030051, China; zhangxi@nuc.edu.cn; 3School of Electronic Information Engineering, Taiyuan University of Science and Technology, Taiyuan 030024, China; zhangdebiao@tyust.edu.cn

**Keywords:** RSSINS, rotation modulation, incomplete modulation error, optimal modulation angular velocity

## Abstract

Single-axis rotation modulation (SRM) still accumulates errors in the roll axis direction, which leads to the navigation accuracy not meeting the requirements of guided missiles. Compound rotation modulation (CRM) superimposes one-dimensional rotation on the basis of SRM, so that the error of the projectile in the direction of the roll axis is also modulated. However, the error suppression effect of CRM is not only affected by the error of the IMU itself, but also related to the modulation angular velocity. In order to improve the accuracy of rotary semi-strapdown inertial navigation system (RSSINS), this paper proposes an optimal rotation angular velocity determination method. Firstly, the residual error in CRM scheme is analyzed; then, the relationship between the incomplete modulation error and the modulation angular velocity in CRM is discussed; finally, a method for determining the optimal modulation angular velocity is proposed (*K*-value method). The analysis of the results shows that the navigation accuracy of the guided projectile is effectively improved with the rotation scheme set at the modulation angular velocity determined by the *K*-value method.

## 1. Introduction

Along with the rapid development of modern weapons, guided artillery shells have fully replaced conventional ammunition due to their advantages of high striking accuracy, and the guidance of conventional ammunition has become a crucial direction for development in the modernization of weapons and equipment in the world. Among them, high-speed rotary munitions have become an important part of precision-guided artillery shells because of their high stability and high information update rate [1]. The strapdown inertial navigation system (SINS) is widely used on various aircraft, ships, and artillery shells because of its high autonomy advantage [2]. However, the error of SINS accumulates over time during the solving process, and it can no longer meet the accuracy requirements of guided artillery shells [3]. How to effectively suppress the errors of sensors has become an important research direction for the precision guidance of artillery shells [4]. In recent years, numerous solutions have been proposed.

Among them, the redundant sensor-based random error measurement scheme and the RSSINS-based sensor constant error compensation scheme have significantly improved the navigation accuracy of guided artillery shells [5,6,7,8]. At the same time, considering the high impact and high rotary velocity of the shell, MEMS sensors have been widely used. However, with the rapid development of high-precision navigation fields, MEMS sensors have larger error defects compared with optical sensors, laser sensors, and other high-precision sensors [9]. According to research, guided projectiles can be discharged at speeds of up to 30 r/s, and the maximum acceleration of guided projectile discharge can reach over 104 ${m/s}^{2}$ [10,11,12]. However, since the solution method is still based on the method of the SINS, the improved SSINS still does not completely eliminate the effect of sensor errors [13,14,15,16].

RSSINS provide a new idea to solve this problem. Rotation modulation can be classified as SRM, CRM, or three-axis rotational modulation [17,18,19,20]. The single-axis rotational modulation technique does not change the internal structure of the system, but only changes the error transmission form of sensors by rotating the IMU, modulating the constant error into the form of a combination of sine and cosine signals. Error integrates to zero over an integer number of cycles, suppressing error divergence [21]. CRM changes the way that the IMU originally makes a single-axis rotation motion around the roll axis and rotates around the roll axis while rotating around the other axis perpendicular to the roll axis, so that the constant error in the direction of the unmodulated roll axis is also modulated. The navigation solution results show that the roll angle accuracy is improved by about 90%, and the position accuracy is improved by 70% [22]. In this context, it is required to complete the fast compensation of errors during the flight of shells with short flight time, such that traditional data processing methods such as Wiener filter and particle filter are not applicable due to their long update period [23,24,25]. Moreover, the angular velocity of the rotating platform also has an effect on the error modulation [26].

Based on CRM, this paper investigates the law of rotational modulation angular velocity in error propagation and proposes a design method for optimal modulation angular velocity by establishing a navigation solution error model in a highly dynamic ballistic environment. The results show that this method can effectively eliminate the bias error in all three axes of the IMU, compensating the part of the sensor output information caused by the bias error and suppressing the error divergence of navigation parameters. The rotation schemes based on the proposed method have generally improved the attitude angle accuracy by 50% and the position accuracy by two orders of magnitude in the navigation settlement results compared with the rest of the rotation schemes.

This paper is organized as follows: in Section 2, the principles of SRM and CRM are briefly introduced, and the residual error model of CRM is established. In Section 3, based on the residual error model obtained from the analysis, the rotational modulation incomplete error equation is derived and an optimal angular velocity determination method that can effectively suppress the modulation incomplete error is proposed, which is named as the *K*-value method. In Section 4, the performance of the proposed method is verified by simulations and experiments. A summary is made in Section 5.

## 2. Principle of RSSINS

### 2.1. Condition of Complete Modulation

The compound RSSINS adds a one-dimensional rotation motion to the uniaxial rotation modulation, which enables the IMU to rotate at a constant angular velocity $\omega_{r1}$ around its roll axis $\left(OY_{b}\right)$ and at a constant angular velocity $\omega_{r2}$ around the other axis perpendicular to the roll axis $\left(OX_{S1}\right)$, but it is still inside the projectile body and follows the projectile. This ensures that the calculation results of navigation parameters can accurately reflect the motion state of the missile. At the same time, in the modulation process, the constant errors of IMU are transformed into the form of the sine-cosine component of the b-coordinate system through a series of coordinate transformations, eliminating the influence of the constant errors on navigation accuracy. The schematic diagram of the structure of the compound RSSINS is shown in Figure 1.

The IMU rotation frame (*S*2-frame) is defined as $\left(OX_{S2}Y_{S2}Z_{S2}\right)$, where *S*2-frame firstly rotates around $\left(OX_{S2}\right)$ axis to the *S*1-frame, which is represented by $C_{S2}^{S1}$. Then, *S*1-frame rotates around axis to the *b*-frame, which is represented by $C_{S1}^{b}$. The relative positional relationship of the three frames of the *b*-frame, the *S*1-frame, and the *S*2-frame is shown in Figure 1. The matrix expression for the transformation of the *S*2-frame to the *b*-frame at moment *t* is shown in Equation (1).
(1)$$C_{S2}^{b}=C_{S1}^{b}C_{S2}^{S1}=\left[\begin{array}{ccc} \cos\omega_{r1}t & \sin\omega_{r1}t\sin\omega_{r2}t & \sin\omega_{r1}\cos\omega_{r2}t\\ 0 & \cos\omega_{r2}t & -\sin\omega_{r2}t\\ -\sin\omega_{r1}t & \cos\omega_{r1}t\sin\omega_{r2}t & \cos\omega_{r1}t\cos\omega_{r2}t \end{array}\right]$$

Let $a=\omega_{r1}+\omega_{r2};\;b=\omega_{r1}-\omega_{r2}$, then the matrix can be equivalent to Equation (2).
(2)$$C_{S2}^{b}=\left[\begin{array}{ccc} \cos\omega_{r1}t & \frac{\cos at-\cos bt}{2} & \frac{\sin at+\sin bt}{2}\\ 0 & \cos\omega_{r2}t & -\sin\omega_{r2}t\\ -\sin\omega_{r1}t & \frac{\sin at-\sin bt}{2} & \frac{\cos at+\cos bt}{2} \end{array}\right]$$

For the convenience of the research problem, we assume that the navigation frame (*n*-frame) and the carrier frame (*b*-frame) are coincident, which represents $C_{b}^{n}=I$. At moment *t*, the bias error of the gyro is coupled to its output, and the modulation in the *n*-frame takes the form shown in Equation (3).
(3)$$\delta \omega_{iS2}^{n}=\varepsilon^{n}=C_{b}^{n}C_{S2}^{b}\varepsilon^{S2}=\left[\begin{array}{c} \varepsilon_{x}^{S2}\cos\omega_{r1}t-\frac{\varepsilon_{y}^{S2}}{2}(\cos at-\cos bt)+\frac{\varepsilon_{z}^{S2}}{2}(\sin at+\sin bt)\\ \varepsilon_{y}^{S2}\cos\omega_{r2}t-\varepsilon_{z}^{S2}\sin\omega_{r2}t\\ -\varepsilon_{x}^{S2}\sin\omega_{r1}t+\frac{\varepsilon_{y}^{S2}}{2}(\sin at-\sin bt)+\frac{\varepsilon_{z}^{S2}}{2}(\cos at+\cos bt) \end{array}\right]$$
where $\varepsilon^{S2}={\left[\begin{array}{ccc} \varepsilon_{x}^{S2} & \varepsilon_{y}^{S2} & \varepsilon_{z}^{S2} \end{array}\right]}^{T}$ is the bias error of the MEMS gyroscope in the *S*2-frame. $\delta \omega_{iS2}^{n}$ is the angular velocity error caused by the gyroscope bias error in the *n*-frame. The analysis shows that bias error of the gyroscope is no longer a normal value in CRM but is modulated as a combination of sine and cosine components of fixed frequency in the *n*-frame. When the modulation period of CRM is the lowest common multiple of each component modulation period, each component can be eliminated. The period when the bias error of gyroscope in three directions is accumulated to zero is defined as *T*. The expression of *T* is shown in Equation (4):
(4)$$T=\left[\begin{array}{ccc} 2\pi /\omega_{r1}, & 2\pi /\omega_{r2}, & \begin{array}{cc} 2\pi /a, & 2\pi /b \end{array} \end{array}\right]$$

The form of bias error modulation of the accelerometer is the same as that of the gyroscope. Considering the influence of the two modulation angular velocities introduced into the navigation scheme on the coordinate transformation matrix, the navigation solution process of the compound rotation modulation scheme is shown in Figure 2.

As shown in Figure 2, $\omega_{r1}$ is the modulation angular velocity of the rotating platform A and $\omega_{r2}$ is the modulation angular velocity of the rotating platform B. When IMU is sensitive to the change of motion state, it outputs angular velocity and specific force information. After coordinate changing, the signal in *S*2-frame is converted to *n*-frame, and then the navigation parameters are calculated.

Through observation, *a* and *b* have a significant effect on the error suppression effect as modulation frequencies. When $\omega_{r1}$ and $\omega_{r2}$ take the same value, the modulation frequency b goes to zero in ideal state, resulting in invalid results of the modulation period solving. For working out this problem, we analyze the error modulation form of CRM when a and b are respectively zeroed to determine the value range of $\omega_{r1}$ and $\omega_{r2}$. Assuming $\omega_{r1}=\omega_{r2}$ and substituting it into the above equation, the modulation form of the bias error in *n*-frame is shown in Equation (5).
(5)$$\delta \omega_{iS2}^{b}(\omega_{r1}=\omega_{r2})=\left[\begin{array}{c} \varepsilon_{x}^{S2}\cos\omega_{r1}t+\frac{\varepsilon_{y}^{S2}}{2}\cos2\omega_{r1}t+\frac{\varepsilon_{z}^{S2}}{2}\sin\omega_{r1}t-\frac{\varepsilon_{y}^{S2}}{2}+\frac{\varepsilon_{z}^{S2}}{2}\\ \varepsilon_{y}^{S2}\cos\omega_{r1}t-\varepsilon_{z}^{S2}\sin\omega_{r1}t\\ -\varepsilon_{x}^{S2}\sin\omega_{r1}t+\frac{\varepsilon_{y}^{S2}}{2}\sin2\omega_{r1}t+\frac{\varepsilon_{z}^{S2}}{2}\cos2\omega_{r1}t-\frac{\varepsilon_{y}^{S2}}{2}+\frac{\varepsilon_{z}^{S2}}{2} \end{array}\right]$$

It can be inferred from the above equation that when the modulation angular velocities of the rotating platforms A and B are the same, there is still a constant value independent of the periodic component in the error modulation result, and the bias error cannot be fully modulated. Similarly, in the case where the two modulation angular velocities are the same, but the rotation directions are opposite ($\omega_{r1}=-\omega_{r2}$), the constant error cannot be fully modulated either. Under the condition of $\omega_{r1}\neq \pm \omega_{r2}$, the integral result of bias error in the whole period T is shown in Equation (6).
(6)$$\int_{0}^{T}\delta \omega_{iS2}^{n}(\omega_{r1}\neq \pm \omega_{r2})=\int_{0}^{T}\left[\begin{array}{ccc} \cos\omega_{r1}t & \frac{\cos at-\cos bt}{2} & \frac{\sin at+\sin bt}{2}\\ 0 & \cos\omega_{r2}t & -\sin\omega_{r2}t\\ -\sin\omega_{r1}t & \frac{\sin at-\sin bt}{2} & \frac{\cos at+\cos bt}{2} \end{array}\right]\left[\begin{array}{c} \varepsilon_{x}^{S2}\\ \varepsilon_{y}^{S2}\\ \varepsilon_{z}^{S2} \end{array}\right]=\left[\begin{array}{c} 0\\ 0\\ 0 \end{array}\right]$$

From the above equation, the condition that CRM can completely eliminate the influence of bias error is shown in Equation (7).
(7)$$\left(\omega_{r1}\neq \pm \omega_{r2}\right)\wedge \left(\omega_{r1}\neq 0\right)\wedge \left(\omega_{r2}\neq 0\right)$$

### 2.2. Residual Error Model of Compound RSSINS

#### 2.2.1. Bias Error

The modulation form of bias error based on CRM has been given in Equation (3).

#### 2.2.2. Scale Factor Error

The error transmission law of the scale factor under the new scheme is analyzed to discern whether the new scheme introduces extra errors. We assume that the carrier is placed stationary on platform B at a pitch angle of 90°. The rotating platform A rotates around the $Y_{b}$ axis at the angular velocity $\omega_{r1}$ and the rotating platform B rotates around the $X_{S1}$ axis at the angular velocity $\omega_{r2}$. Then the angular velocity output of the three-axes gyroscope without scale factor error coupling state is shown in Equation (8).
(8)$$\omega_{iS2}^{S2}={\left[\begin{array}{ccc} -\omega_{r2} & -\omega_{r1}\cos\omega_{r2}t & \omega_{r1}\sin\omega_{r2}t \end{array}\right]}^{T}$$

The expression of the scale factor error coupled to the gyroscope output under the compound rotation modulation scheme, in the *b*-frame, is shown in Equation (9).
(9)$$\delta \omega_{iS2,S}^{b}=\left[\begin{array}{ccc} \delta S_{g,x} & 0 & 0\\ 0 & \delta S_{g,y} & 0\\ 0 & 0 & \delta S_{g,z} \end{array}\right]\left[\begin{array}{c} -\omega_{r2}\\ -\omega_{r1}\cos\omega_{r2}t\\ \omega_{r1}\sin\omega_{r2}t \end{array}\right]=\left[\begin{array}{c} -\delta S_{g,x}\omega_{r2}\\ -\delta S_{g,y}\omega_{r1}\cos\omega_{r2}t\\ \delta S_{g,z}\omega_{r1}\sin\omega_{r2}t \end{array}\right]$$

In *n*-frame, the gyroscope output angular velocity error caused by the scale factor error is shown in Equation (10).


(10)
$$\begin{array}{l} \delta \omega_{iS2,S}^{n}=\left[\begin{array}{ccc} \cos\omega_{r1}t & \sin\omega_{r1}t\sin\omega_{r2}t & \sin\omega_{r1}\cos\omega_{r2}t\\ 0 & \cos\omega_{r2}t & -\sin\omega_{r2}t\\ -\sin\omega_{r1}t & \cos\omega_{r1}t\sin\omega_{r2}t & \cos\omega_{r1}t\cos\omega_{r2}t \end{array}\right]\left[\begin{array}{c} -\delta S_{g,x}\omega_{r2}\\ -\delta S_{g,y}\omega_{r1}\cos\omega_{r2}t\\ \delta S_{g,z}\omega_{r1}\sin\omega_{r2}t \end{array}\right]\\ =\left[\begin{array}{c} -\delta S_{g,x}\omega_{r2}\sin\omega_{r1}t+\frac{\delta S_{g,y}-\delta S_{g,z}}{4}\omega_{r1}\left[\cos(\omega_{r1}+2\omega_{r2})t-\cos(\omega_{r1}-2\omega_{r2})t\right]\\ -\frac{\delta S_{g,y}-\delta S_{g,z}}{2}\omega_{r1}\cos2\omega_{r2}t-\frac{\delta S_{g,y}+\delta S_{g,z}}{2}\omega_{r1}\\ \delta S_{g,x}\omega_{r2}\sin\omega_{r1}t-\frac{\omega_{r1}(\delta S_{g,y}-\delta S_{g,z})}{4}[\sin(\omega_{r1}+2\omega_{r2})t-\sin(\omega_{r1}-2\omega_{r2})] \end{array}\right] \end{array}$$


According to the basic principle of rotation modulation, $T_{1}$ is the maximum common multiple of each sine and cosine component period, and $T_{1}$ is defined as the period in which the scalar factor error of IMU is completely modulated to zero, whose expression is shown in Equation (11).
(11)$$T_{1}=\left[\begin{array}{cccc} 2\pi /\omega_{r1}, & 2\pi /(\omega_{r1}-2\omega_{r2}), & 2\pi /\omega_{r2}, & 2\pi /(\omega_{r1}+2\omega_{r2}) \end{array}\right]$$

Similarly, discussing the condition of $\omega_{r1}=2\omega_{r2}$, the modulation form of the scale factor error is shown in Equation (12).
(12)$$\delta \omega_{iS2,S}^{n}(\omega_{r1}=2\omega_{r2})=\left[\begin{array}{c} -\delta S_{g,x}\omega_{r2}\cos2\omega_{r2}t+\frac{\delta S_{g,y}-\delta S_{g,z}}{4}\omega_{r2}\left[\cos(4\omega_{r2})t-1\right]\\ -\frac{\delta S_{g,y}-\delta S_{g,z}}{2}\omega_{r1}\cos2\omega_{r2}t-\frac{\delta S_{g,y}+\delta S_{g,z}}{2}\omega_{r2}\\ \delta S_{g,x}\omega_{r2}\sin\omega_{r2}t-\frac{\omega_{r2}(\delta S_{g,y}-\delta S_{g,z})}{4}\sin4\omega_{r2}t \end{array}\right]$$

From the above equation, it can be seen that under the condition of $\omega_{r1}=2\omega_{r2}$, there is still a constant component in the modulation result of the scalar factor error, and the scalar factor error cannot be completely modulated. In CRM, the output angular velocity error excited by the gyroscope scale factor error is obtained after the integration of the whole cycle $T_{1}$ and it is shown in Equation (13).
(13)$$\delta \theta_{S}(\omega_{r1}=2\omega_{r2})=\int_{0}^{T_{1}}\delta \omega_{iS2,S}^{n}(\omega_{r1}=2\omega_{r2})dt={\left[\begin{array}{ccc} \frac{\delta S_{g,y}-\delta S_{g,z}}{2}\omega_{r2}T & 0 & 0 \end{array}\right]}^{T}$$

Under the condition of $\omega_{r1}\neq 2\omega_{r2}$, the output angular velocity error excited by the gyroscope scale factor error is obtained after the integration of the whole cycle $T_{1}$ as shown in Equation (14).
(14)$$\delta \theta_{S}(\omega_{r1}\neq 2\omega_{r2})=\int_{0}^{T_{1}}\delta \omega_{iS2,S}^{n}(\omega_{r1}\neq 2\omega_{r2})dt={\left[\begin{array}{ccc} 0 & 0 & 0 \end{array}\right]}^{T}$$

From the above analysis, it can be seen that the modulation angular velocity condition that can satisfy the angular velocity error $\left(\delta \theta_{S}\right)$ caused by the scale factor error accumulated to zero in one complete cycle is shown in Equation (15). We can draw the conclusion that CRM does not introduce a new scale factor error term compared with SRM.
(15)$$\left(\omega_{r1}\neq \pm 2\omega_{r2}\right)\wedge \left(\omega_{r1}\neq 0\right)\wedge \left(\omega_{r2}\neq 0\right)$$

#### 2.2.3. Installation Error

In CRM, the propagation law of installation error is analyzed to discern whether a new error term is introduced in this scheme. Consistent with the conditions used in the analysis of the scale factor error, the installation error coupled to the gyroscope output in CRM is expressed in the form as shown in Equation (16).
(16)$$\begin{array}{l} \delta \omega_{iS2,N}^{S2}=\left[\begin{array}{ccc} 0 & \delta G_{z} & -\delta G_{y}\\ -\delta G_{z} & 0 & \delta G_{x}\\ \delta G_{y} & -\delta G_{x} & 0 \end{array}\right]\left[\begin{array}{c} -\omega_{r2}\\ -\omega_{r1}\cos\omega_{r2}t\\ \omega_{r1}\sin\omega_{r2}t \end{array}\right]\\ =\left[\begin{array}{c} -\delta G_{z}\omega_{r1}\cos\omega_{r2}t-\delta G_{y}\omega_{r1}\sin\omega_{r2}t\\ \delta G_{z}\omega_{r2}+\delta G_{x}\omega_{r1}\sin\omega_{r2}t\\ -\delta G_{y}\omega_{r2}+\delta G_{x}\omega_{r1}\cos\omega_{r2}t \end{array}\right] \end{array}$$

When carrier is at stationary state, in the *n*-frame, the angular velocity error of the gyroscope output caused by the installation error is shown in Equation (17).
(17)$$\delta \omega_{iS2,N}^{n}=\left[\begin{array}{c} \left(\begin{array}{l} -\frac{\delta G_{z}}{2}(\omega_{r1}(\cos at+\cos bt)+\omega_{r2}(\cos at-\cos bt))\\ -\frac{\delta G_{y}}{2}(\omega_{r1}(\sin at-\sin bt)+\omega_{r2}(\sin at+\sin bt))+\delta G_{x}\omega_{r1}\sin\omega_{r1}t \end{array}\right)\\ \delta G_{z}\omega_{r2}\cos\omega_{r2}t+\delta G_{y}\omega_{r2}\sin\omega_{r2}t\\ \left(\begin{array}{l} \frac{\delta G_{z}}{2}(\omega_{r1}(\sin at+\sin bt)+\omega_{r2}(\sin at-\sin bt))\\ -\frac{\delta G_{y}}{2}(\omega_{r1}(\cos at-\cos bt)+\omega_{r2}(\cos at+\cos bt))+\delta G_{x}\omega_{r1}\sin\omega_{r1}t \end{array}\right) \end{array}\right]$$

According to the basic principle of rotation modulation, $T_{2}$ is the maximum common multiple of each sine and cosine component period, and $T_{2}$ is defined as the period in which the installation factor error of IMU is completely modulated to zero, whose expression is shown in Equation (18).
(18)$$T_{2}=\left[\begin{array}{cccc} 2\pi /\omega_{r1}, & 2\pi /(\omega_{r1}+\omega_{r2}), & 2\pi /\omega_{r2}, & 2\pi /(-\omega_{r1}+\omega_{r2}) \end{array}\right]$$

Analyzing the frequency of each error term, the gyroscope output angular velocity error caused by the installation error is shown in Equation (19).


(19)
$$\begin{array}{l} \delta \omega_{iS2,N}^{n}(\omega_{r1}=\omega_{r2})=\left[\begin{array}{c} \left(\begin{array}{l} -\frac{\delta G_{z,g}}{2}(\omega_{r2}(\cos2\omega_{r2}t+1)+\omega_{r2}(\cos2\omega_{r2}t-1))\\ -\delta G_{y,g}\omega_{r2}\sin2\omega_{r2}t+\delta G_{x,g}\omega_{r2}\sin\omega_{r2}t \end{array}\right)\\ \delta G_{z,g}\omega_{r2}\cos\omega_{r2}t+\delta G_{y,g}\omega_{r2}\sin\omega_{r2}t\\ \left(\begin{array}{l} \delta G_{z,g}\omega_{r2}\sin2\omega_{r2}t-\frac{\delta G_{y,g}}{2}(\omega_{r2}(\cos2\omega_{r2}t-1)\\ +\omega_{r2}(\cos2\omega_{r2}t+1))+\delta G_{x,g}\omega_{r2}\sin\omega_{r2}t \end{array}\right) \end{array}\right]\\ =\left[\begin{array}{c} \left(\begin{array}{l} -\delta G_{z,g}\omega_{r2}\cos2\omega_{r2}t-\delta G_{y,g}\omega_{r2}\sin2\omega_{r2}t\\ +\delta G_{x,g}\omega_{r2}\sin\omega_{r2}t) \end{array}\right)\\ \delta G_{z,g}\omega_{r2}\cos\omega_{r2}t+\delta G_{y,g}\omega_{r2}\sin\omega_{r2}t\\ \left(\begin{array}{l} \delta G_{z,g}\omega_{r2}\sin2\omega_{r2}t-\delta G_{y,g}\omega_{r2}\cos2\omega_{r2}t\\ +\delta G_{x,g}\omega_{r2}\sin\omega_{r2}t \end{array}\right) \end{array}\right] \end{array}$$


Under the condition of $\omega_{r1}=\omega_{r2}$ or $\omega_{r1}=-\omega_{r2}$, the modulation result of the installation error still contains a constant component, and the installation error cannot be completely modulated. In CRM, the output angular velocity error excited by the gyroscope installation error is integrated over the whole period $T_{2}$ to obtain the angular error as shown in Equation (20).
(20)$$\delta \theta_{N}(\omega_{r1}=\omega_{r2})=\int_{0}^{T_{2}}\delta \omega_{iS2,N}^{n}(\omega_{r1}=\omega_{r2})dt={\left[\begin{array}{ccc} -\delta G_{z,g}\frac{b}{2}T_{2} & 0 & \delta G_{y,g} \end{array}\frac{a}{2}T_{2}\right]}^{T}$$

As can be seen from the above equation, the installation error accumulates in the $X_{n}$ axis and $Z_{n}$ axis directions after the whole-cycle integration of the gyro output angular velocity error, and similarly the accelerometer specific force output error caused by the installation error under the condition of $\omega_{r1}=-\omega_{r2}$ cannot be completely modulated. Under the condition of $\omega_{r1}\neq \pm \omega_{r2}$, at the whole cycle duration of $T_{2}$, the angular error obtained by integrating the gyroscope output error caused by the installation error is shown in Equation (21).
(21)$$\delta \theta_{N}(\omega_{r1}\neq \pm \omega_{r2})=\int_{0}^{T_{2}}\delta \omega_{iS2,N}^{n}(\omega_{r1}\neq \omega_{r2})dt={\left[\begin{array}{ccc} 0 & 0 & 0 \end{array}\right]}^{T}$$

There is no constant component of the gyroscope output error caused by the installation error, and the installation error is completely modulated. From the above analysis, it can be seen that, ideally, the modulation angular velocity condition that can satisfy the angular error $\delta \theta_{N}$ caused by the installation error in a complete cycle without constant accumulation can be shown by Equation (22).
(22)$$\left(\omega_{r1}\neq \pm \omega_{r2}\right)\wedge \left(\omega_{r1}\neq 0\right)\wedge \left(\omega_{r2}\neq 0\right)$$

Therefore, the installation error under the compound rotational modulation scheme does not excite the new sensors output error.

### 2.3. Error Transfer Model for Compound RSSINS

Due to the low accuracy of MEMS sensors, the information of the Earth’s self-propagating angular velocity and the change of the Earth’s surface curvature cannot be sensitized. Therefore, the magnitude of $\omega_{in}^{n}=\omega_{ie}^{n}+\omega_{en}^{n}$ is small. The influence from these factors can be ignored in the RSSINS solving process. Meanwhile, the coordinate transformation matrix becomes complicated due to the introduction of an additional rotating platform, and the propagation form of errors is also changed in the process of solving navigation parameters. In CRM, the deviation angle error is shown in Equation (23).
(23)$$\dot{\phi }=\phi \times \omega_{in}^{n}+\delta \omega_{in}^{n}-C_{b}^{n}C_{S1}^{b}C_{S2}^{s1}(K_{g}\omega_{iS2}^{S2}+\varepsilon^{S2})$$
where ${\left[\begin{array}{ccc} \phi_{E} & \phi_{N} & \phi_{U} \end{array}\right]}^{T}$ are the deviation angle errors in the east, north and sky directions, respectively. $\omega_{in}^{n}$ is the rotational angular velocity of the *n*-frame relative to the *i*-frame, $\omega_{iS2}^{S2}$ is the projection of the gyroscope output on the *S*2-frame, and $\varepsilon^{S2}$ is the gyroscope bias error. The simplified posture angle error transfer model is shown in Equation (24) and the velocity error equation for CRM is shown in Equation (25).
(24)$$\dot{\phi }=-C_{b}^{n}C_{S1}^{b}C_{S2}^{S1}(K_{g}\omega_{iS2}^{S2}+\varepsilon^{S2})$$
(25)$$\delta {\dot{V}}^{n}=\phi^{n}\times f_{iS2}^{n}+C_{b}^{n}C_{S1}^{b}C_{S2}^{S1}(K_{a}f^{S2}+\nabla^{S2})-\delta V^{n}\times (2\omega_{ie}^{n}+\omega_{en}^{n})+V^{n}\times (2\delta \omega_{ie}^{n}+\delta \omega_{en}^{n})$$
where $V^{n}=\left[\begin{array}{ccc} V_{E} & V_{N} & V_{U} \end{array}\right]$ is the velocity in the east, north, and sky directions. $\delta {\dot{V}}^{n}$ is the velocity error in the three directions, and $f_{iS2}^{S2}$ is the specific force to which the accelerometer is sensitive in the *S*2-frame. Similarly, due to the fact that the MEMS gyroscope is not sensitive to the angular velocity of the Earth’s rotation and the small rotation of the *n*-frame caused by the curvature of the Earth’s surface, the order of magnitude of $\delta V^{n}\times (2\omega_{ie}^{n}+\omega_{en}^{n})+V^{n}\times (2\delta \omega_{ie}^{n}+\delta \omega_{en}^{n})$ is small and negligible. The simplified velocity error transfer model is shown in Equation (26).
(26)$$\delta {\dot{V}}^{n}=\phi^{n}\times f_{sf}^{n}+C_{b}^{n}C_{S1}^{b}C_{S2}^{S1}(K_{a}f^{S2}+\nabla^{S2})$$

The position error equation under CRM is shown in Equation (27).
(27)$$\left\{\begin{array}{c} \delta \dot{L}=\frac{\delta V_{N}}{R_{M}+h}-\delta h\frac{V_{N}}{{\left(R_{M}+h\right)}^{2}}\\ \begin{array}{l} \delta \dot{\lambda }=\frac{\delta V_{E}}{R_{N}+h}\sec L+\delta L\frac{\delta V_{E}}{R_{N}+h}\tan L\sec L-\delta h\frac{V_{E}\sec L}{{\left(R_{N}+h\right)}^{2}} \end{array}\\ \delta h=\delta V_{U} \end{array}\right.$$
where $\left[\begin{array}{ccc} \delta L & \delta \lambda & \delta h \end{array}\right]$ is the latitude, longitude, and altitude errors, respectively, and $\left[\begin{array}{ccc} \delta V_{E} & \delta V_{N} & \delta V_{U} \end{array}\right]$ is the velocity error in the eastward, northward, and skyward directions, respectively. The above equation shows that the accelerometer error is transferred to the position error in the form of quadratic integration, which takes the form of a quadratic function in the position error, while the gyroscope error is transferred to the position error in the form of cubic integration. The velocity of error accumulation accelerates as time grows. When the carrier is stationary or moving a short distance, the formula of position error can be equivalent to Equation (28).
(28)$$\delta \dot{P}=\delta V$$

## 3. Optimal Rotation Angular Velocity Determination Method (*K*-Value Method)

### 3.1. Modulation Incomplete Error of RSSINS

#### 3.1.1. The Modulation Incomplete Error of Angular Velocity

Rotation modulation technique successfully achieves error suppression by modulating the error as a periodic signal, which in turn modulates the initially linearly increasing error with time to zero in a single integration operation. However, the existing method has two problems in principle: firstly, since the effective navigation time of the carrier is short in the missile environment; according to previous studies, the minimum modulation period of CRM should satisfy the requirement of the period of each component. This leads to a situation wherein one flight process may not cover one rotation modulation period, causing a lag in information update; second, the error modulation accumulates to zero after one integration, while the signal depends on the motion state of the carrier. After the second or higher integration, the error will no longer be a periodic signal symmetric about y = 0, but propagates in a higher-order form, and the error has a residual term in the integration operation of the integer period. The carrier is at stationary state and the deviation angle error at moment *t* is shown in Equation (29).
(29)$$\phi (t)=-\int_{0}^{t}C_{b}^{n}C_{S1}^{b}C_{S2}^{S1}(\delta K_{g}\omega_{iS2}^{S2}+\varepsilon^{S2})dt=-\int_{0}^{t}\delta \omega_{iS2,\varepsilon }^{n}dt-\int_{0}^{t}\delta \omega_{iS2,S}^{n}dt-\int_{0}^{t}\delta \omega_{iS2,N}^{n}dt$$

The deviation angle error caused by each error of the gyroscope is calculated separately for the three integrals as shown in Equation (30).
(30)$$\phi (t)=\left[\begin{array}{c} \phi_{1}(t)-\frac{\varepsilon_{z}^{S2}}{ab}\omega_{r1}\\ \phi_{2}(t)+\frac{\varepsilon_{z}^{S2}}{\omega_{r2}}+\frac{\delta S_{g,y}\omega_{r1}}{4\omega_{r2}}-\delta G_{y}\\ \phi_{3}(t)+\frac{\varepsilon_{x}^{S2}}{\omega_{r1}}+\frac{\varepsilon_{y}^{S2}\omega_{r2}}{ab}-\frac{\omega_{r2}}{\omega_{r1}}\delta S_{g,x}+\frac{\omega_{r1}\omega_{r2}(\delta S_{g,y}-\delta S_{g,z})}{a^{2}-b^{2}}-\delta G_{z} \end{array}\right]$$ where $\left[\begin{array}{ccc} \phi_{1}(t) & \phi_{2}(t) & \phi_{3}(t) \end{array}\right]$ are symmetric sine and cosine periodic functions about y = 0, which integrate to zero in a complete cycle, and $\left[\begin{array}{cccc} \omega_{r1} & \omega_{r2} & a & b \end{array}\right]$ is fixed under the determination scheme. The mean value of the deviation angle error in the whole cycle is constant, while the installation error and the scale factor error can be ignored due to the magnitude being small. The mean value of deviation angle error is shown in Equation (31).
(31)$$E[\phi (t)]={\left[\begin{array}{ccc} -\frac{\omega_{r1}\varepsilon_{z}^{S2}}{ab} & \frac{\varepsilon_{z}^{S2}}{\omega_{r2}} & -\frac{\varepsilon_{x}^{S2}}{\omega_{r1}}-\frac{\varepsilon_{y}^{S2}\omega_{r2}}{ab} \end{array}\right]}^{T}$$

It can be seen from the above analysis that although the rotation modulation technique suppresses the divergence of the bias error, it still causes the deviation angle error in principle. Under ideal conditions, there is still an error that fluctuates around a constant value in the deviation angle error. Through comparison, it can be seen that CRM introduces a larger deviation angle error amplitude than SRM, and it is introduced into the attitude calculation error in the subsequent calculation. Meanwhile, the deviation angle error caused by incomplete modulation error will be transferred to the velocity error in the form of $\delta V_{\phi }^{n}=\int_{0}^{T}\phi \times f^{n}dt$. Then, the deviation angle error caused by incomplete modulation angular velocity will cause the divergence of the velocity error. When the carrier is at rest, the velocity error due to deviation angle error is as shown in Equation (32).
(32)$$\left\{\begin{array}{c} \delta V_{E}=\int_{0}^{T}\left(-\phi_{U}f_{N}+\phi_{N}f_{U}\right)dt=-(\frac{\varepsilon_{x}^{S2}}{\omega_{r1}}+\frac{\varepsilon_{y}^{S2}\omega_{r2}}{ab})f_{N}T+\frac{\varepsilon_{z}^{S2}}{\omega_{r2}}f_{U}T\\ \delta V_{N}=\int_{0}^{T}\left(\phi_{U}f_{E}-\phi_{E}f_{U}\right)dt=(\frac{\varepsilon_{x}^{S2}}{\omega_{r1}}+\frac{\varepsilon_{y}^{S2}\omega_{r2}}{ab})f_{E}T+\frac{\varepsilon_{z}^{S2}\omega_{r1}}{ab}f_{U}T\\ \delta V_{U}=\int_{0}^{T}\left(-\phi_{N}f_{E}+\phi_{E}f_{N}\right)dt=-\varepsilon_{z}^{S2}T(\frac{f_{E}}{\omega_{r2}}+\frac{\omega_{r1}f_{N}}{ab}) \end{array}\right.$$

We can conclude that an incomplete modulation angular velocity still exists in the principle of CRM. There is a certain deviation angle error, which acts on the attitude solution, so that the deviation angle angular error is modulated as a superposition of a constant error and a periodically varying error, and the deviation angle also diverges over time. On the other hand, it acts on the velocity solution so that the velocity errors in the east and north directions are diverged.

#### 3.1.2. The Modulation Incomplete Error of Acceleration

In addition to the velocity error component from the deviation angle error, the bias error of the accelerometer also transfers to the velocity and position errors through the integration operation. However, the accelerometer error propagation mechanism is more complex, and more errors are excited under static conditions. At moment *t*, the velocity error component $\delta V_{\nabla }^{n}$ caused by the bias error of the accelerometer is shown in Equation (33).


(33)
$$\begin{array}{l} \delta V_{\nabla }^{n}(t)=\int_{0}^{t}\delta f_{iS2,\nabla }^{n}dt\\ =\left[\begin{array}{c} \frac{\nabla_{z}^{S2}}{ab}\omega_{r1}+\frac{\nabla_{x}^{S2}}{\omega_{r1}}\sin\omega_{r1}t+\frac{\nabla_{y}^{S2}}{2}(\frac{\sin bt}{b}-\frac{\sin at}{a})-\frac{\nabla_{z}^{S2}}{2}(\frac{\cos bt}{a}+\frac{\cos bt}{b})\\ \frac{\nabla_{y}^{S2}}{\omega_{r2}}\sin\omega_{r2}t+\frac{\nabla_{z}^{S2}}{\omega_{r2}}\cos\omega_{r2}t-\frac{\nabla_{z}^{S2}}{\omega_{r2}}\\ -\frac{\nabla_{x}^{S2}}{\omega_{r1}}-\frac{\nabla_{y}^{S2}}{ab}\omega_{r2}+\frac{\nabla_{x}^{S2}}{\omega_{r1}}\cos\omega_{r1}t-\frac{\nabla_{y}^{S2}}{2}(\frac{\cos at}{a}-\frac{\cos bt}{b})+\frac{\nabla_{z}^{S2}}{2}(\frac{\sin at}{a}+\frac{\sin bt}{b}) \end{array}\right] \end{array}$$


The mean error over a whole period is shown in Equation (34).
(34)$$E[\delta V_{\nabla }^{n}]={\left[\begin{array}{ccc} \frac{\nabla_{z}^{S2}}{ab}\omega_{r1} & -\frac{\nabla_{z}^{S2}}{\omega_{r2}} & -\frac{\nabla_{x}^{S2}}{\omega_{r1}}-\frac{\nabla_{y}^{S2}}{ab}\omega_{r2} \end{array}\right]}^{T}$$

At moment *t*, the position error component $\delta P_{\nabla }^{n}$ caused by the bias error of the accelerometer is shown in Equation (35).


(35)
$$\begin{array}{l} \delta P_{\nabla }^{n}(t)=\int_{0}^{t}\delta V_{\nabla }^{n}(t)dt\\ =\left[\begin{array}{c} \left(\begin{array}{l} \frac{\nabla_{x}^{S2}}{\omega_{r1}{}^{2}}+\frac{\omega_{r1}\nabla_{z}^{S2}}{ab}t-\frac{\nabla_{y}^{S2}}{2}(\frac{1}{a^{2}}-\frac{1}{b^{2}})-\frac{\nabla_{x}^{S2}}{\omega_{r1}{}^{2}}\cos\omega_{r1}t\\ +\frac{\nabla_{y}^{S2}}{2}(\frac{\cos at}{a^{2}}-\frac{\cos bt}{b^{2}})-\frac{\nabla_{z}^{S2}}{2}(\frac{\sin at}{a^{2}}+\frac{\sin bt}{b^{2}}) \end{array}\right)\\ \frac{\nabla_{y}^{S2}}{\omega_{r2}{}^{2}}-\frac{\nabla_{z}^{S2}}{\omega_{r2}}t-\frac{1}{\omega_{r2}{}^{2}}(\nabla_{y}^{S2}\cos\omega_{r2}t-\nabla_{z}^{S2}\sin\omega_{r2}t)\\ \left(\begin{array}{l} -\frac{\nabla_{x}^{S2}}{\omega_{r1}}t-\frac{\omega_{r2}\nabla_{y}^{S2}}{ab}t+\frac{\nabla_{z}^{S2}}{2}(\frac{1}{a^{2}}+\frac{1}{b^{2}})+\frac{\nabla_{x}^{S2}\sin\omega_{r1}t}{\omega_{r1}{}^{2}}\\ -\frac{\nabla_{y}^{S2}}{2}(\frac{\sin at}{a^{2}}-\frac{\sin bt}{b^{2}})-\frac{\nabla_{z}^{S2}}{2}(\frac{\cos at}{a^{2}}+\frac{\cos bt}{b^{2}}) \end{array}\right) \end{array}\right] \end{array}$$


The cumulative increment of $\delta P_{\nabla }^{n}$ after a complete cycle is shown in Equation (36).
(36)$$\delta P_{\nabla }^{n}(T)-\delta P_{\nabla }^{n}(0)={\left[\begin{array}{ccc} \frac{\omega_{r1}\nabla_{z}^{S2}}{ab}T & -\frac{\nabla_{z}^{S2}}{\omega_{r2}}T & -\frac{\nabla_{x}^{S2}}{\omega_{r1}}T-\frac{\omega_{r2}\nabla_{y}^{S2}}{ab}T \end{array}\right]}^{T}$$

In the *n*-frame, the specific force error $\delta f_{iS2,S}^{n}$ in the output of the accelerometer due to the scale factor error is shown in Equation (37).


(37)
$$\begin{array}{l} \delta f_{iS2,S}^{n}=C_{S2}^{b}\delta f_{iS2,S}^{S2}\\ =\left[\begin{array}{c} \left(\begin{array}{l} -\frac{2\delta S_{a,x}-\delta S_{a,z}}{4}(\sin2\omega_{r1}t)-\frac{\delta S_{a,y}}{8}(\cos2at-\cos2bt)\\ +\frac{\delta S_{a,z}}{8}(\sin2at+\sin2bt) \end{array}\right)\\ \frac{\delta S_{a,y}-\delta S_{a,z}}{4}(\sin(\omega_{r1}+2\omega_{r2})t-\sin(\omega_{r1}-2\omega_{r2})t)\\ \left(\begin{array}{l} \frac{\delta S_{a,x}}{2}+\frac{\delta S_{a,y}+\delta S_{a,z}}{4}-\frac{\cos2\omega_{r1}}{2}(\delta S_{a,x}-\frac{\delta S_{a,y}+\delta S_{a,z}}{2})\\ -\frac{\delta S_{a,y}-\delta S_{a,z}}{4}(\cos2\omega_{r2}t+\cos2at+\cos2bt) \end{array}\right) \end{array}\right] \end{array}$$


At moment *t*, the velocity error $\delta V_{S}^{n}$ caused by the scale factor error is shown in Equation (38). The mean value of the velocity error $\delta V_{S}^{n}$ over a whole period is shown in Equation (39).


(38)
$$\begin{array}{l} \delta V_{S}^{n}(t)=\int_{0}^{t}\delta f_{iS2,S}^{n}dt\\ =\left[\begin{array}{c} \left(\begin{array}{l} \frac{\delta S_{a,z}}{16}(\frac{1}{a}+\frac{1}{b})-\frac{2\delta S_{a,x}-\delta S_{a,z}}{8\omega_{r1}}+\frac{2\delta S_{a,x}-\delta S_{a,z}}{8\omega_{r1}}\cos2\omega_{r1}t\\ -\frac{\delta S_{a,y}}{16}(\frac{\sin2at}{a}+\frac{\sin2bt}{b})-\frac{\delta S_{a,z}}{16}(\frac{\cos2at}{a}+\frac{\cos2bt}{b}) \end{array}\right)\\ \frac{\delta S_{a,y}-\delta S_{a,z}}{4}(-\frac{\cos(\omega_{r1}+2\omega_{r2})t}{\omega_{r1}+2\omega_{r2}}+\frac{\cos(\omega_{r1}-2\omega_{r2})t}{\omega_{r1}-2\omega_{r2}})-\frac{\omega_{r2}}{\omega_{r1}{}^{2}-4\omega_{r2}{}^{2}}(\delta S_{a,y}-\delta S_{a,z})\\ \left(\begin{array}{l} \frac{\delta S_{a,x}}{2}t+\frac{\delta S_{a,y}+\delta S_{a,z}}{4}t-\frac{\sin2\omega_{r1}t}{4\omega_{r1}}(\delta S_{a,x}-\frac{\delta S_{a,y}+\delta S_{a,z}}{2})\\ -\frac{\delta S_{a,y}-\delta S_{a,z}}{8}(\frac{\sin2\omega_{r2}t}{\omega_{r2}}+\frac{\sin2at}{a}+\frac{\sin2bt}{b}) \end{array}\right) \end{array}\right] \end{array}$$



(39)
$$E[\delta V_{S}^{n}]=\left[\begin{array}{c} \frac{\delta S_{a,z}}{16}(\frac{1}{a}+\frac{1}{b})-\frac{2\delta S_{a,x}-\delta S_{a,z}}{8\omega_{r1}}\\ -\frac{\omega_{r2}}{\omega_{r1}{}^{2}-4\omega_{r2}{}^{2}}(\delta S_{a,y}-\delta S_{a,z})\\ \frac{\delta S_{a,x}}{2}t+\frac{\delta S_{a,y}+\delta S_{a,z}}{4}t \end{array}\right]$$


It is known that the cumulative increment of the position error $\delta P_{S}^{n}$ caused by the scale factor error in a complete period is shown in Equation (40).
(40)$$\delta P_{S}^{n}(T)-\delta P_{S}^{n}(0)=\left[\begin{array}{c} \frac{\delta S_{a,z}}{16}(\frac{1}{a}+\frac{1}{b})T-\frac{2\delta S_{a,x}-\delta S_{a,z}}{8\omega_{r1}}T\\ -\frac{\omega_{r2}}{\omega_{r1}{}^{2}-4\omega_{r2}{}^{2}}(\delta S_{a,y}-\delta S_{a,z})T\\ \frac{\delta S_{a,x}}{2}T^{2}+\frac{\delta S_{a,y}+\delta S_{a,z}}{4}T^{2} \end{array}\right]$$

From the analysis, it can be seen that the incomplete modulation error caused by the scale factor error is small in magnitude and can be ignored, and the incomplete modulation error caused by the installation error can be ignored in the same way.

### 3.2. Optimal Rotation Angle Velocity Determination Method

Through the above study, we can find that two rotation modulation angular velocities of the rotating platform A and B are the main factors affecting the deviation angle error, velocity error, and position error. The addition of one-dimensional rotational motion to CRM leads to a more variable arrangement of the two modulation angular velocities. After scientific analysis, the ratio of two modulation angular velocities is taken as the independent variable affecting the error, and the trend of the navigation error is observed by changing the magnitude of the ratio. This method is named as the *K*-value method. The *K*-value method is defined as the proportional relationship between two modulation angular velocities which represents $K=\omega_{r2}/\omega_{r1}$. The feasible conditions for CRM are obtained in the previous study as shown in Equation (41).
(41)$$\left(\omega_{r1}\neq 0\right)\wedge \left(\omega_{r2}\neq 0\right)\wedge \left(\omega_{r1}\neq \pm \omega_{r2}\right)\wedge \left(\omega_{r1}\neq \pm 2\omega_{r2}\right)$$

After derivation, the position error increment $\delta P^{n}$ caused by incomplete modulation error over the complete cycle is shown in Equation (42).
(42)$$\left\{\begin{array}{c} \delta P_{\varepsilon }^{n}=\left[\begin{array}{ccc} \frac{\varepsilon_{z}^{S2}}{\omega_{r2}}T^{2} & \frac{\omega_{r1}\varepsilon_{z}^{S2}}{ab}T^{2} & 0 \end{array}\right]\\ \delta P_{\nabla }^{n}=\left[\begin{array}{ccc} \frac{\omega_{r1}\nabla_{z}^{S2}}{ab}T & -\frac{\nabla_{z}^{S2}}{\omega_{r2}}T & -\frac{\nabla_{x}^{S2}}{\omega_{r1}}T-\frac{\omega_{r2}\nabla_{y}^{S2}}{ab}T \end{array}\right] \end{array}\right.$$

Substituting $K=\omega_{r2}/\omega_{r1}$ into Equation (42), we can obtain the relationship between *K* and the position error.
(43)$$\left\{\begin{array}{c} \delta P_{\varepsilon }^{n}=\left[\begin{array}{ccc} \frac{T^{2}\varepsilon_{z}^{S2}}{\omega_{r1}K} & \frac{T^{2}\varepsilon_{z}^{S2}}{\omega_{r1}(1-K^{2})} & 0 \end{array}\right]\\ \delta P_{\nabla }^{n}=\left[\begin{array}{ccc} \frac{T\nabla_{z}^{S2}}{\omega_{r1}(1-K^{2})} & -\frac{T\nabla_{z}^{S2}}{\omega_{r1}K} & -\frac{T}{\omega_{r1}}(\nabla_{x}^{S2}+\frac{K\nabla_{y}^{S2}}{1-K^{2}}) \end{array}\right] \end{array}\right.$$

Through the analysis, we can find that when the rotation modulation angular velocity $\omega_{r1}$ of the rotating platform A is determined, the only factor that affects the position error is the value of *K*. In the following, the optimal rotation modulation angle velocity is determined by analyzing the trend of the accumulated increment of the position error when the value of *K* changes.

By observing the position error increments in the eastward, northward, and skyward directions as the value of *K* changes, we can find that the skyward position error increments caused by incomplete modulation error is the largest when *K* takes the value of ±1; the northward and eastward position errors increments caused by incomplete modulation error are the largest when *K* takes the values of ±1 and 0. These three rotation schemes have been proven to be infeasible. When the value range of *K* is $\left[-1,1\right]$, the error increment caused by incomplete modulation error is large. Therefore, the rotation scheme with the value of *K* in $\left[-1,1\right]$ is still not the optimal rotation scheme. When the value range of *K* is more than 1 or less than −1, the error increment caused by the change of *K* needs to be further studied through the error change trend of navigation parameters.

## 4. Simulation and Experimentation

### 4.1. Rotation Schemes for Different K

By designing different arrangements of *K*, the motion state of the projectile in different environments was simulated, and the accuracy of the navigation solution of different schemes was compared. It was verified that the optimal rotation modulation angle velocity determination method based on CRM in this paper has the best error suppression effect. The simulated IMU output data was used to compare the error suppression effect between different rotation modulation angular velocity scheduling methods in each environment. The error parameters of the IMU are shown in Table 1. According to the device accuracy of commonly used low- and medium-precision MEMS sensors, we designed the bias errors of gyro and accelerometer as 24 °/s and 2 mg respectively, while scale factor errors were both 50 ppm, and installation errors were both $5^{\prime }$.

Based on the set IMU parameters, the output information of angular velocity and specific force were generated. In order to verify the correctness of the theory that *K* is the only influencing factor affecting the error divergence trend, eight sets of rotation schemes with different *K* were designed. The simulation results were observed to analyze whether the error dispersion trend is consistent with the results of the theoretical analysis as the *K* changes. The rotation scheme corresponding to the highest navigation accuracy was selected. The designed rotation schemes with different *K* values are shown in Table 2.

At the same time, the error suppression effect of different *K* schemes in the above table is discussed under three different motion states.

#### 4.1.1. Stationary State

For the system in the stationary state, eight groups of experiments were carried out according to the above rotation scheme to analyze the variation law of errors. Figure 3a shows the comparison of the deviation angle in the eastward, northward, and skyward directions for the eight sets of experiments with different values of *K*. Figure 3b shows the comparison of the velocity errors in the eastward, northward, and skyward directions for the eight sets of experiments with different values of *K*.

It is obvious from Figure 3 that the deviation angle and velocity errors are modulated into periodic functions due to the effect of CRM, while the northward errors diverge over time under the condition of SRM (*K* = 0). As shown in the figures, the deviation angle in the eastward, northward, and skyward directions diverge when *K* = 1. *K* = 0 is the single-axis rotation modulation scheme, which is consistent with the previous analysis, and the deviation angle in the northward direction (roll axis direction) diverges under this scheme, while the deviation angle error in eastward direction and skyward directions is suppressed significantly. Based on the above figures, we can conclude that the error divergence is serious under the two modulation schemes of *K* = 0 and *K* = 1 when the carrier is at stationary state and the remaining modulation schemes have similar error suppression effects.

#### 4.1.2. Yaw Motion State

In the yaw motion environment, the different schemes in the table were used for modulation, and the error suppression performances of different schemes under the dynamic state of angular motion were compared. The carrier motion states are set in Table 3. In order to study the influence of different *K* schemes on navigation solution accuracy in yaw motion state, we designed a scheme with carrier turns at an angular velocity of 2 °/s and 3 °/s and observed the deviation angle and velocity error of the carrier.

The error-free ideal trajectory generated by the simulation is shown in Figure 4. Since the set trajectory requires the carrier to turn around in the yaw motion mode, and the trajectory finally returns to the starting point (the starting point is represented by dots in the figure), the ideal motion trajectory matches the design scheme.

For the system in the state of yaw motion, eight groups of experiments were carried out according to the above rotation scheme to analyze the variation law of errors.

Figure 5a shows the comparison of the deviation angle in the eastward, northward, and skyward directions for the eight sets of experiments with different values of *K*. Figure 5b shows the comparison of the velocity errors in the eastward, northward, and skyward directions for the eight sets of experiments with different values of *K*. As shown in the figures, the deviation angle and velocity errors diverge seriously under the two rotation schemes of *K* = 0 and *K* = 1, which are consistent with the simulation results of the carrier at stationary state. the deviation angle and velocity error have a smaller degree of dispersion under different *K*. The error fluctuations are more obvious when the motion state changes abruptly.

#### 4.1.3. Acceleration and Deceleration Motion State

The different schemes in the table were modulated to compare the error suppression performance of the different schemes in the acceleration and deceleration dynamic state. The carrier motion states are set in Table 4.

For the system in the acceleration and deceleration motion state, six groups of experiments were carried out according to the above rotation scheme to analyze the variation law of errors.

Figure 6a shows the comparison of the deviation angle in the eastward, northward, and skyward directions for the eight sets of experiments with different values of *K*. Figure 6b shows the comparison of the velocity errors in the eastward, northward, and skyward directions for the eight sets of experiments with different values of *K*. As shown in the figures, the two rotation schemes, *K* = 0 and *K* = 1, have been proven to be ineffective rotation schemes and need not be discussed further. It is necessary to analyze the error divergence of the rotation modulation scheme in the case of sudden acceleration and deceleration motion: due to the linear acceleration and deceleration of the carrier, only the eastward velocity error fluctuates up and down with the carrier motion state, while the northward and skyward velocity errors still disperse according to the original rule. Under the condition of *K* = 1/2, the skyward deviation angle divergence is serious, and the northward velocity error diverges in the scheme of *K* = 4. In particular, the deviation angle and velocity error suppression effects of the two schemes, *K* = 2 and *K* = 3, are similar.

In these three states of motion, the simulation results of different rotation schemes were analyzed, and we can conclude that when *K* is taken in the interval of $\left[-1,1\right]$, the eastward deviation angle and velocity error dispersion have strong uncertainty and the modulation angular speed should avoid *K* = 0 and *K* = 1; considering the stability of the motor, modulation angular velocity should be as small as possible to reduce angular velocity output error of the motor.

Meanwhile, after simulating and comparing the error dispersion results under *K* = 2 and *K* = 3 conditions, *K* = 2 and *K* = 3 were chosen to design the rotation modulation scheme: $\omega_{r1}=\;60\;{}^{\circ}/s$**,**
$\omega_{r2}=\;120\;{}^{\circ}/s$; $\omega_{r1}=\;30\;{}^{\circ}/s$, $\omega_{r2}=\;90\;{}^{\circ}/s$. However, since the modulation periods under the *K* = 2 and *K* = 3 schemes are 6 s and 12 s, respectively, and the error suppression effect of the two schemes is similar, the rotation modulation scheme with *K* = 2 was chosen: $\omega_{r1}=\;60\;{}^{\circ}/s$**,**
$\omega_{r2}=\;120\;{}^{\circ}/s$.

### 4.2. Rotation Schemes for the Same K

To verify the effect of numerical value of the modulation angular velocity on error suppression effect when the system is at stationary state, the rotation scheme with the same *K* was chosen. According to the above study, the modulation effect of deviation angle and velocity error is best when *K* = 2. Eight sets of rotation schemes with different modulation angular velocities but the same *K* were selected, and the modulation angular velocities were arranged as shown in Table 5.

The IMU data generated by the set rotation schemes are shown in Figure 7a and the angular velocity and specific force output under the *S*2-frame are shown in Figure 7b for Schemes 4 and 5.

As can be seen from Figure 7, the output information of sensors is modulated into a periodic signal due to CRM. For the system in the stationary state, eight groups of experiments were carried out according to the above rotation schemes to analyze the variation law of errors.

Figure 8a shows the comparison of deviation angle in the eight groups of experiments with the same *K* in the eastward, northward, and skyward directions. Figure 8b shows the comparison of velocity errors in eastward, northward, and skyward directions in eight groups of experiments with the same *K*. As shown in the figures, the modulation period decreases from Schemes 1–8 when the ratio of the two modulation angular velocities is a certain value, which means that the larger the modulation angular velocity is, the shorter the modulation period is. Schemes 1 and 2 have a longer modulation period, and the amplitude of the modulated error in the complete period is large, which cannot meet the update rate of effective navigation information carried by short-duration missiles.

At the same time, it can be concluded that the larger the modulation angular velocity is, the smaller the amplitude of the modulation error in the complete period is. Under the condition that the modulation angle velocity is more than 20 °/s, the navigation performance is similar between different angle velocity schemes when the two modulation angular velocities *K* are certain. Therefore, the key to determine the optimal modulation angular velocity lies in the determined *K*.

### 4.3. Simulation of Rotation Scheme in Missile Environment

The rotation angular velocities of platforms were simulated in missile environment and set according to the different *K*-value schemes. The navigation solution results were analyzed to verify that the error suppression is best at the rotation modulation angular velocities determined by the *K*-value method. The ballistic simulation program was used to simulate the motion of the carrier in missile environment and generate the output information of the inertial sensors during the flight. According to the set rotation schemes, the simulation results were observed and the error suppression effect of the rotation modulation was analyzed. The indexes of the designed projectile are shown in Table 6.

Ballistic simulation experiments were conducted to verify that the optimal rotation modulation angular velocity scheme determined by the *K*-value method has the best navigation accuracy under the condition of set IMU parameters. The simulated error-free IMU output information is shown in Figure 9, during the flight of the missile. The trajectory comparison diagram of SINS, SRM, CRM, and the trajectory under ideal conditions are shown in Figure 10. As shown in Figure 9a, the rolling axis of the missile was selected as the *X*-axis. The acceleration value is the largest at the moment of its discharge, which is about 9 ${m/s}^{2}$. In the process of smooth flight, since the projectile is almost parallel to the horizontal plane, there is no gravitational acceleration component in the direction of the roll axis, which is about 0. During the descent of the projectile, the gravitational acceleration acts on the roll axis, resulting in a temporary increase in the acceleration of the *X*-axis. As shown in Figure 9b, the roll angle velocity of the missile is the largest at the moment of its discharge, and the roll angle velocity decreases linearly at the stage of steady flight and descent of the missile. The whole flight lasts about 47 s.

Figure 11a–c show the comparison of the missile’s attitude error, velocity error, and position error, respectively, for different schemes. It can be seen that under the modulation scheme determined by the *K*-value method, all errors of CRM maintain good convergence characteristics. In particular, the position error of CRM is one order of magnitude lower than that of SRM and sins. At the same time, we can note that the errors of SRM in the eastward and skyward are less than SINS, which proves the partial modulation effect of SRM.

Table 7 shows the maximum values of each error for the two schemes, the single-axis rotation scheme, and the compound rotation modulation scheme determined by the optimal rotation modulation angular velocity method. Analyzing the table, we can conclude that CRM has the best error suppression effect, and the errors of both SRM and INS have different degrees of divergence. Among them, the roll angle error, eastward velocity error, and eastward position error of SRM are extremely close to those of SINS, reflecting the defect that SRM cannot modulate the error in the direction of the rotation axis of the projectile. For the improvement of attitude accuracy, CRM improves the accuracy of roll angle by about 70% and pitch angle and yaw angle by about 50%. For the improvement of position accuracy, CRM is about two orders of magnitude better than SRM. It is therefore verified that the optimal rotation modulation angular velocity scheme significantly improves the navigation accuracy of the projectile in the missile environment.

To sum up, this paper proposes an optimal rotary modulation angle velocity determination method, which will significantly improve the current situation of serious error divergence in INS and thus improve the accuracy of navigation solution. In the future, the precise control of rotating platform will be another important direction for this research.

## 5. Conclusions

In this paper, based on CRM, a method to determine the optimal rotation modulation angular velocity is proposed. This method not only meets the condition of high update rate of information, but also eliminates the incomplete modulation error introduced by the rotating platforms maximally compared with other arrangements of rotation angular velocities. As a result, the navigation accuracy is improved.

By analyzing the simulation results, the proposed *K*-value method can effectively improve the navigation accuracy of high-rotation missiles. The following conclusions can be drawn from this paper:

1.To explore whether the angular velocity of the rotating platforms causes the introduction of an extra error term in CRM, the propagation form of the constant error was re-modeled. The results show that the ratio of the rotational angular velocities of the two rotating platforms is the only factor that affects the accuracy of the navigation results, provided that $\omega_{r1}$ is certain.2.In order to study the influence of *K* changes on error dispersion, CRM with different *K* was designed for three motion states. It can be found that when the value range of *K* is $\left[-1,1\right]$, the error dispersion is serious and there is great uncertainty in the error variation. Under the condition of *K* = 2 and *K* = 3, the best error suppression effect is achieved.3.Six sets of rotational modulation schemes with the same *K* but different rotational angular velocities were designed in order to investigate whether the different rotation modulation angular velocities affect the error suppression effect. Analyzing the simulation results, it can be found that when the modulation angle velocity is lower than 20 °/s, the larger the modulation angular velocity is, the smaller the error amplitude in the complete cycle. When the modulation angular velocity exceeds 20 °/s, the effect of the rotation modulation angular velocity size on the error suppression effect is not obvious under the same *K*.

## Figures and Tables

**Figure 1 sensors-22-04583-f001:**
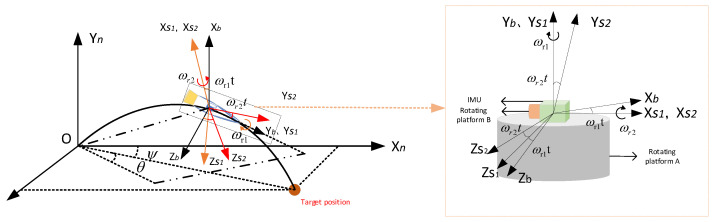
Schematic diagram of the structure of compound RSSINS.

**Figure 2 sensors-22-04583-f002:**
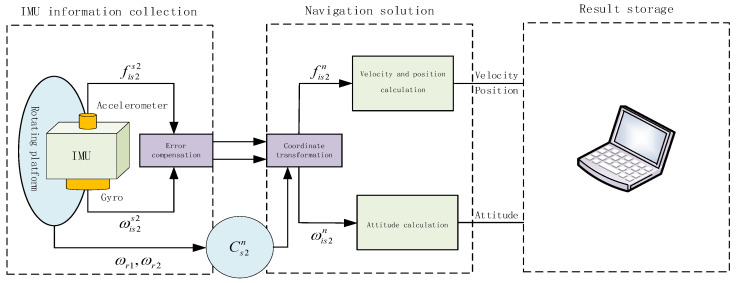
Solution principle block diagram of compound RSSINS.

**Figure 3 sensors-22-04583-f003:**
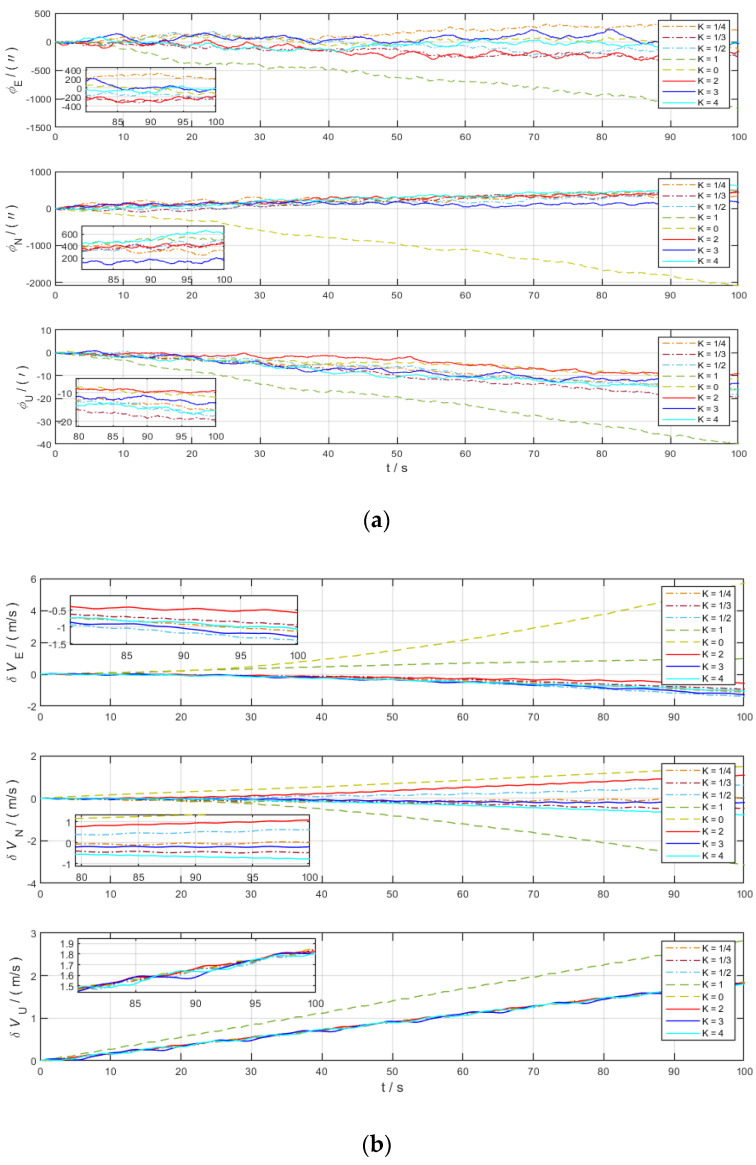
Error comparison chart of different *K* in static state. (**a**) Deviation angle in three axes; (**b**) velocity error in three axes.

**Figure 4 sensors-22-04583-f004:**
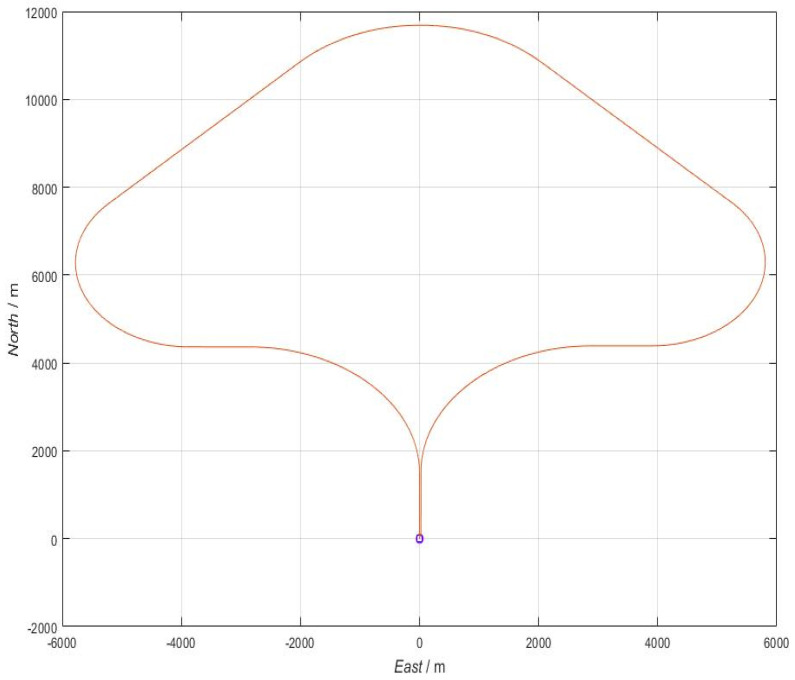
The error-free ideal trajectory in angular motion state.

**Figure 5 sensors-22-04583-f005:**
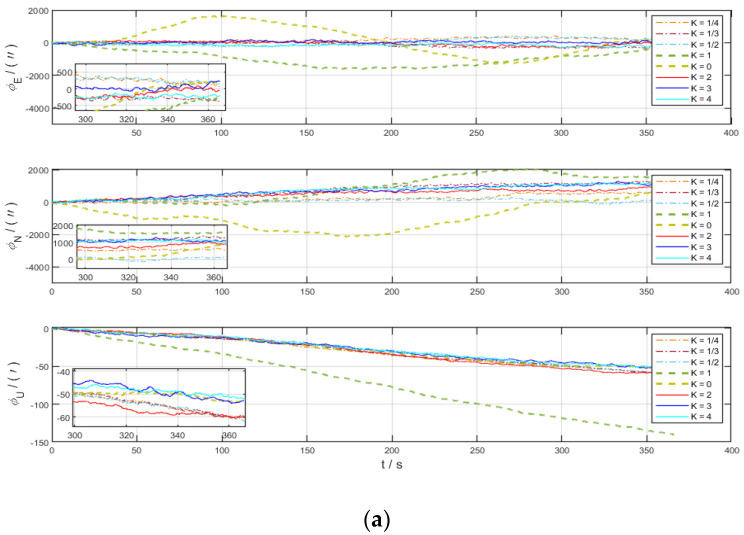
Error comparison chart of different *K* in yaw motion state. (**a**) Deviation angle in three axes; (**b**) velocity error in three axes.

**Figure 6 sensors-22-04583-f006:**
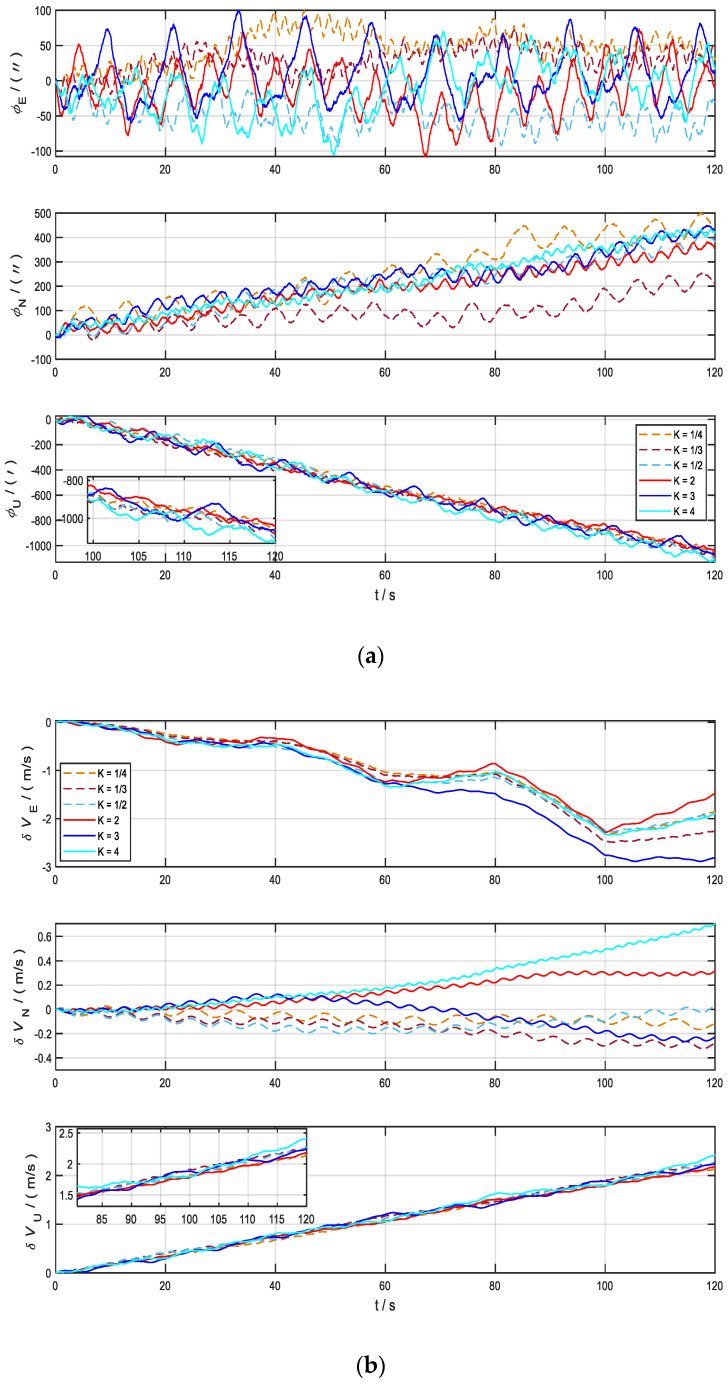
Error comparison chart of different *K* in acceleration and deceleration motion state. (**a**) Deviation angle in three axes; (**b**) velocity error in three axes.

**Figure 7 sensors-22-04583-f007:**
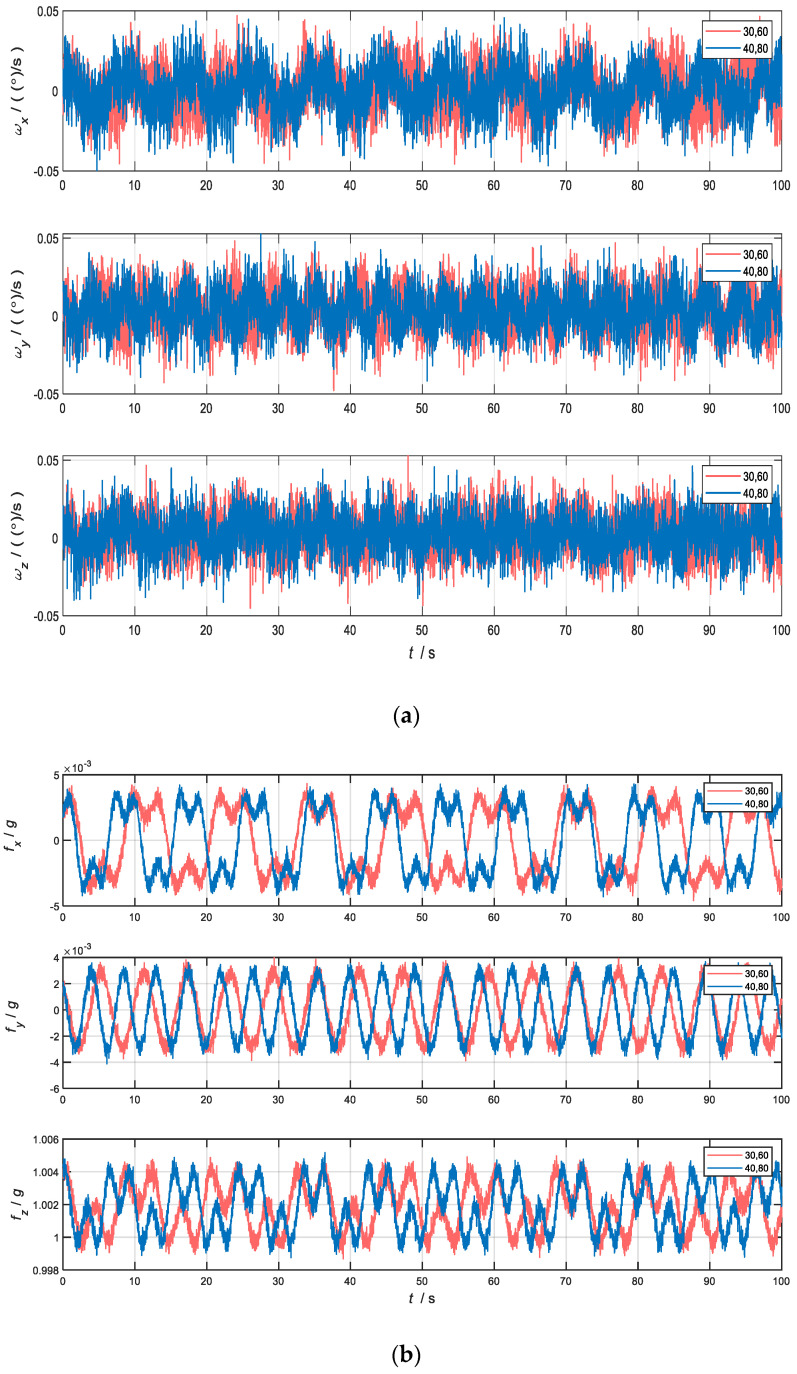
Angular velocity and specific force of the sensors in *b*-frame. (**a**) Angular velocity in three axes; (**b**) specific force in three axes.

**Figure 8 sensors-22-04583-f008:**
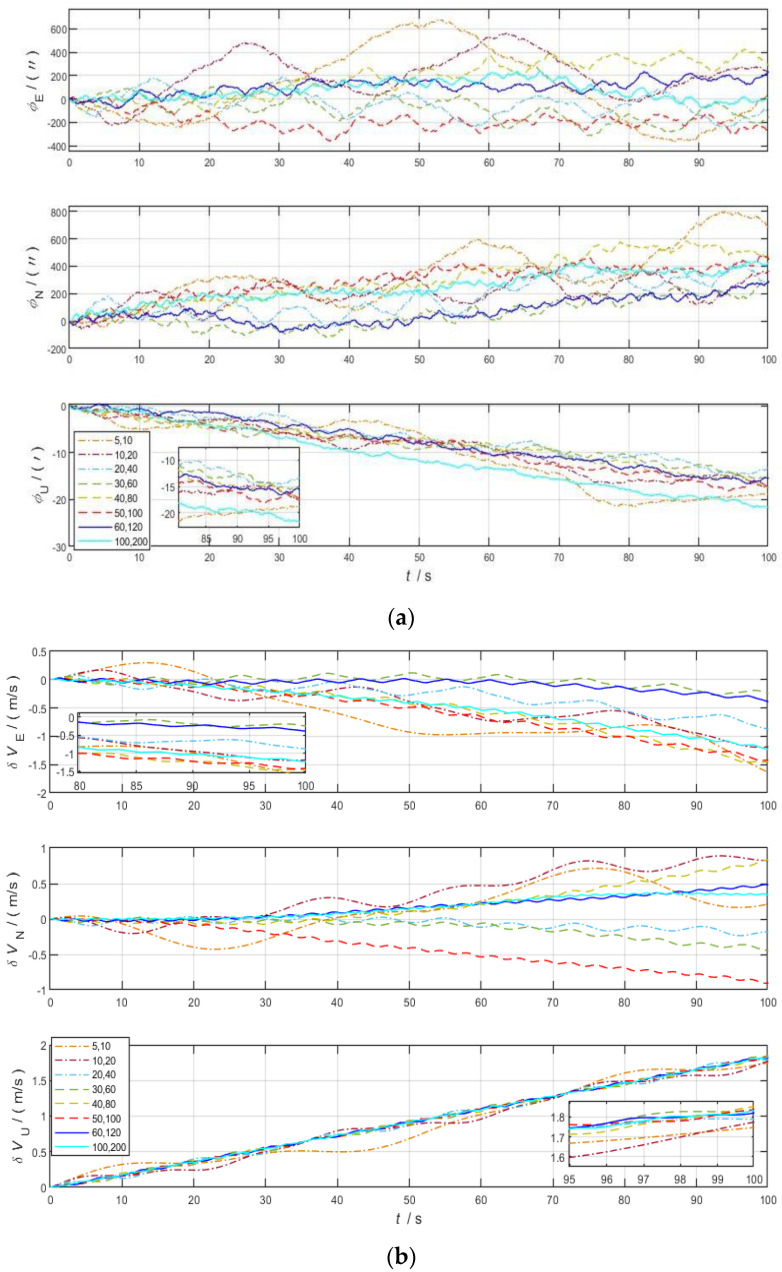
Error comparison chart of the same *K* in stationary state. (**a**) Deviation angle in three axes; (**b**) velocity error in three axes.

**Figure 9 sensors-22-04583-f009:**
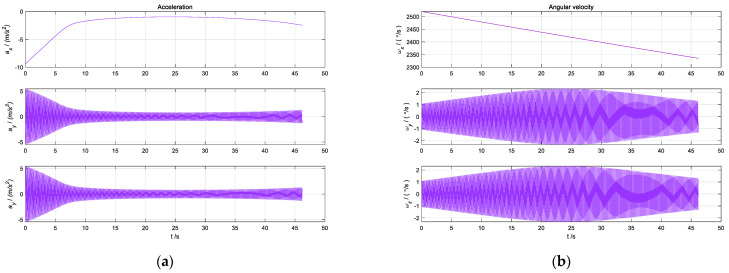
Output information of IMU. (**a**) Output information of angular velocity. (**b**) output information of acceleration.

**Figure 10 sensors-22-04583-f010:**
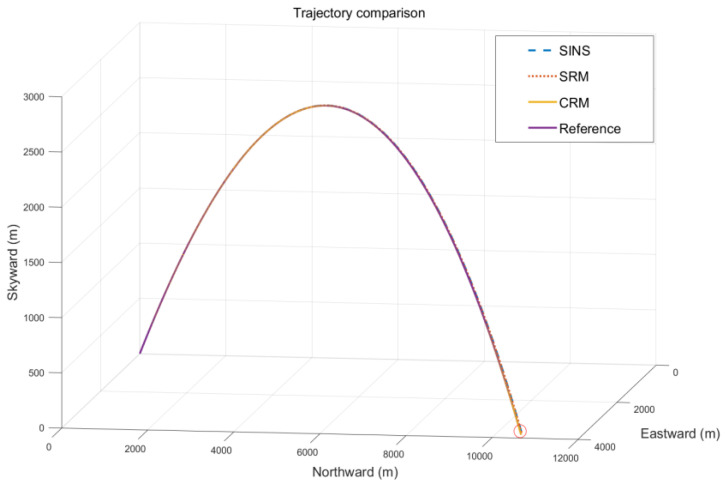
Trajectory comparison chart of different navigation schemes.

**Figure 11 sensors-22-04583-f011:**
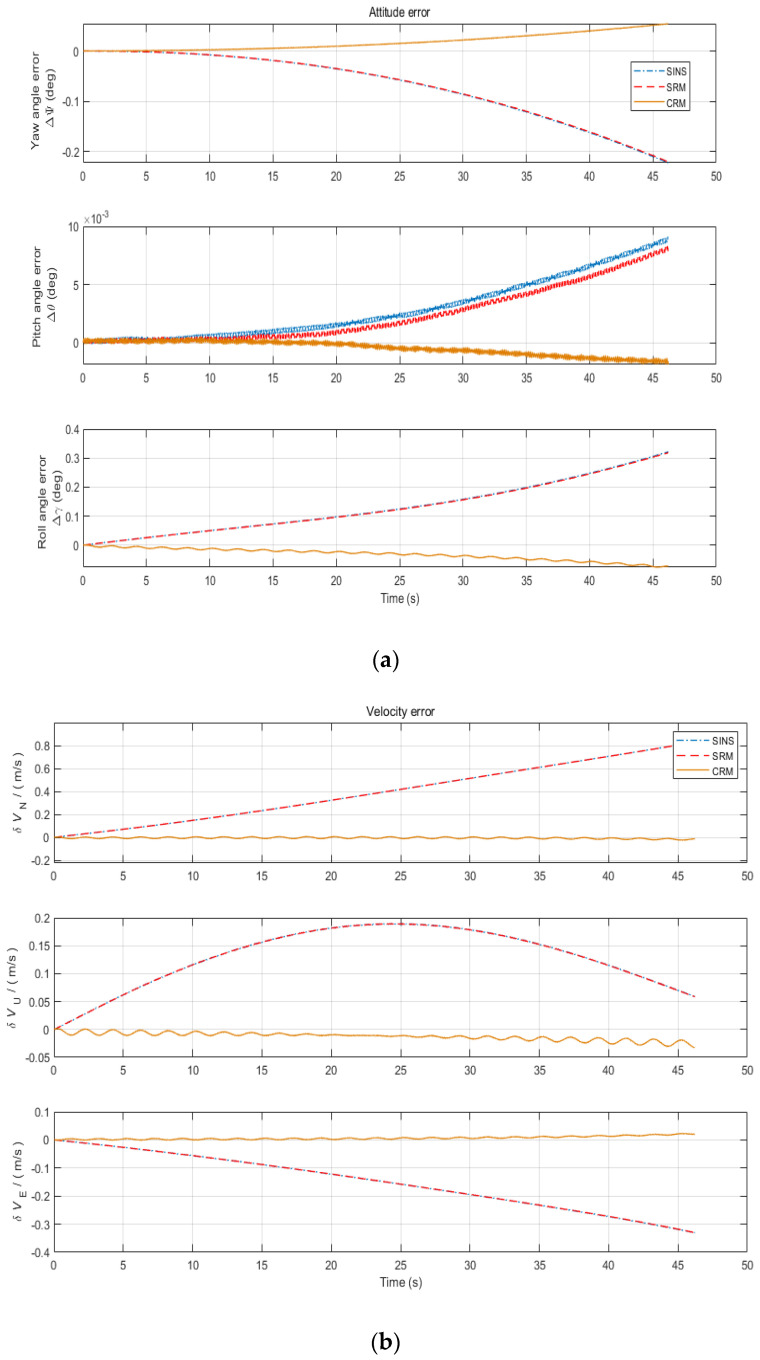
Error comparison chart of different navigation schemes. (**a**) Attitude error comparison chart; (**b**) velocity error comparison chart; (**c**) position error comparison chart.

**Table 1 sensors-22-04583-t001:** IMU error parameter.

MEMS Sensors	Bias Error	Scale Factor Error	Installation Error	Random Wandering
Gyro	24 °/s	50 ppm	$5^{\prime }$	0.28 °/s
Accelerometer	2 mg	50 ppm	$5^{\prime }$	$50\;\mu g/\sqrt{\mathrm{Hz}}$

**Table 2 sensors-22-04583-t002:** Simulation rotation scheme with different *K*.

*K*	$\omega_{r1}$ (°/s)	$\omega_{r2}$ (°/s)
1/4	180	45
1/3	180	60
1/2	120	60
1	120	120
0	120	0
2	60	120
3	60	180
4	45	180

**Table 3 sensors-22-04583-t003:** Setting of angular motion state.

Serial Number	Movement Status	Duration (s)
1	$\mathrm{Accelerate}\;(10\;{m/s}^{2})$	10
2	Turn left (2 °/s)	45
3	Uniform	10
4	Turn right (3 °/s)	45
5	Uniform	45
6	Turn right (2 °/s)	45
7	Uniform	45
8	Turn right (3 °/s)	45
9	Uniform	10
10	Turn left (3 °/s)	45
11	Uniform	10
12	$\mathrm{Decelerate}\;(10\;{m/s}^{2})$	10

**Table 4 sensors-22-04583-t004:** Setting of acceleration and deceleration motion state.

Serial Number	Movement Status	Duration (s)
1	$\mathrm{Accelerate}\;(10\;{m/s}^{2})$	20
2	$\mathrm{Decelerate}\;(10\;{m/s}^{2})$	20
3	$\mathrm{Accelerate}\;(10\;{m/s}^{2})$	20
4	$\mathrm{Decelerate}\;(10\;{m/s}^{2})$	20
5	$\mathrm{Accelerate}\;(10\;{m/s}^{2})$	20
6	$\mathrm{Decelerate}\;(10\;{m/s}^{2})$	20

**Table 5 sensors-22-04583-t005:** The same *K* rotation schemes.

Scheme	*K*	$\omega_{r1}\;({}^{\circ}/s)$	$\omega_{r2}\;({}^{\circ}/s)$
1	2	5	10
2	2	10	20
3	2	20	40
4	2	30	60
5	2	40	80
6	2	50	100
7	2	60	120
8	2	100	200

**Table 6 sensors-22-04583-t006:** The parameters of the simulated missile at the initial moment.

Indicator Items	Numerical Value
Quality	45 kg
Length	1.3 m
Rotational inertia	$0.7\;\mathrm{kg}\cdot m^{2}$
Pneumatic pressure	$105^{\;}\mathrm{kPa}$
Yaw	30°
Pitch	30°
Roll	0°
Latitude	38.1 °N
Longitude	112 °E
Altitude	780 m
Speed	400 m/s
Angular velocity	45 rad/s

**Table 7 sensors-22-04583-t007:** Comparison table of maximum errors of different schemes.

Errors	Single-Axis Rotation Modulation Scheme	Compound Rotation Modulation Scheme
δθ (°)	0.007	−0.003
δγ (°)	0.318	−0.072
δψ (°)	−0.220	0.114
$\delta V_{E}\;(m/s)$	−0.331	0.021
$\delta V_{N}\;(m/s)$	0.829	−0.013
$\delta V_{U}\;(m/s)$	−0.058	−0.032
$\delta P_{E}\;(m)$	−6.917	0.329
$\delta P_{N}\;(m)$	18.119	−0.224
$\delta P_{U}\;(m)$	6.049	−0.581

## Data Availability

Not applicable.

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
