# Peer review of "Optimal Rotational Angular Velocity Determination Method Based on Compound Rotary Semi-Strapdown Inertial Navigation System"

_sensors, 2022, doi:10.3390/s22124583_

Round 1
Reviewer 1 Report
Based on the compound semi-strapdown inertial navigation system, this paper analyzes the influence of the modulation angular velocity of the rotating platform on the navigation accuracy. Referring to the previous research results in inertial navigation and sensor error analysis, this paper puts forward the concept of residual error in the rotation modulation scheme, establishes the transfer model of this error in the navigation solution process, and creatively puts forward the k-value method to determine the optimal modulation angular velocity configuration. First of all, the logic of the article is meticulous and clear, and the mathematical laws are strictly observed in the derivation of formulas. Secondly, this paper starts to study the law in the process of error transmission, the method proposed is innovative and different from the traditional error analysis method. Lastly, this paper has a certain level of scientific research. The method proposed in this paper improves the high-precision navigation ability of inertial navigation system in missile borne environment, and has a certain contribution to the autonomous high-precision strike of missiles. However, the following problems still exist:
1. The working principle of the compound rotary inertial navigation system needs to be introduced more briefly. And the state-of-art methods need to be introduction. The working principle of the rotary modulation system has been mentioned a lot in previous articles. The research focus here is not on the introduction of principle, but on the error suppression method you proposed. The reference, such as Chang L, Qin F, Xu J. Strapdown Inertial Navigation System Initial Alignment Based on Group of Double Direct Spatial Isometries, is the new SINS method. And the reference, which is Gaoge Hu, unscented kalman filter with process noise covariance estimation for vehicular INS/GPS integration system, are the new information fusion method. They are recommended to analysis and cite.
2. Eliminate redundant descriptions in the introduction. As for the research background, the description should be concise and clear, summarizing the research object in this paper and the current development form. It should not be mentioned repeatedly.
3. Modify the content of the keyword. The statement about rotation modulation in the keyword is inconsistent with that in the abstract. Please confirm the concept description in the two places.
4. Modify the abstract content. The introduction of the compound rotary semi-strapdown inertial navigation system in the abstract is too simple, which may cause the concept of modulation angular velocity to become fuzzy, so you should accurately put forward the object you want to study while refining the language
5. The content of line 158 in the body is inconsistent with the context content description. It is recommended to delete it.
6. Simplify derivation formula. The public derivation process should appropriately omit the intermediate steps and only show the important form of the formula. Please delete unimportant or possibly ambiguous formulas.
7. The conclusion part should be more refined to make the findings and contributes of the paper clearer. Furthermore, please note the different between the conclusions and abstract.
8. Revise the conclusion in paragraphs. Conclusions are not just about summarizing the key results of the study; it should highlight the insights and the applicability of your findings for further work. Please make it more concise and show only the high impact outcomes. Report your conclusions in two or maximum three paragraph.
9. Eliminate multiple reference. After that, please check the manuscript thoroughly and eliminate all the lumps in the manuscript. This should be done by characterizing each reference individually. This can be done by mentioning 1 or 2 phrases per reference to show how it is different form the others and why it deserves mentioning.
10. As for the comparison of various errors in the conclusion part, what is your proposed method compared with, and the conclusion that the accuracy is improved is obtained. The improvement of accuracy requires more quantitative data to express, which can not be expressed only by text or graphic results.
Reviewer 2 Report
1) Please explain absolutely all the terms used in the equations, even from the first three equations.
2) Figures 1 and 2 require further discussion.
3) A broader discussion of the statement in lines 128-130 is needed.
4) Explain the sign introduced in lines 146, and 149, between the angular velocities wr1 and wr2.
5) Please check if the comma is still needed at the beginning of equations (13-14) and in line 201, at the lower index, before s?
6) The same question for line 227, equation (20), and line 235, equation (21), and in line 240, about the comma, now put before N.
7) The same remark in line 312 about the comma entered before the FI.
8) The same remark for lines (328, 333, 336, 341, 342, 346) and equations (33, 34, 35, 36, 38, 39, 40, 42, 43).
9) Discuss Tables 1, 2, and 3 in more detail.
10) Discuss Figures 3-11in more detail.
11) A broader discussion of Table 7 is needed.
12) Before concluding, enter a few words about the future research directions opened by the new methodology presented in the paper.
13) Presenting a program used in an appendix would have been a good thing.
Author Response
Please see the attachment.

This manuscript is a resubmission of an earlier submission. The following is a list of the peer review reports and author responses from that submission.